# Recombination alters the receptor binding and furin cleavage site in novel bat-borne HKU5-CoV-2 coronavirus

Ting-Yu Yeh,[1,2] Vincent Tsai,[1,3] Samuel M. Liao,[1,4] Chia-En Hong,[1,5] Feng-Yu Kuo,[1,6] Yen Chun Wang,[1,7] Michael C. Feehley,[1,2,8] Patrick J. Feehley,[1,2,8] Yi-Chen Lai,[9,10] Gregory P. Contreras[1,2]

**ABSTRACT** HKU5-CoV-2 is a new bat-infecting coronavirus phylogenetically related to MERS-CoV. It has recently been confirmed that HKU5-CoV-2 can enter human cells and organoids *in vitro* via the ACE2 receptor, raising concerns about its pandemic potential due to zoonotic spillover. Whether recombination has an influence on HKU5-CoV-2 infectivity or biological fitness is completely unclear to date. Here, we report the first evidence that the HKU5-CoV-2 receptor-binding domain (RBD) and S1/S2 furin cleavage site (FCS) of the spike protein are recombination hotspots. Using linkage disequilibrium and haploblock analysis, we identified 167 recombination breakpoints and 27 haploblocks. SNP23016/23043/23064/23156/23193/23285 at the RBD and SNP23833/23847 at the FCS are recombinant breakpoints. Our results suggest that recombination may lead to the substitution at RBD residue 498 (Thr498Val/Val498Thr and Thr498Ile/Ile498Thr), which Thr498 directly contacts the ACE2 receptor. Recombination also causes Ser723 deletion/insertion and Ser729Ala substitution at the FCS. These mutations could affect host tropism and change furin cleavage activity. Our results indicate that recombination has played a critical role in HKU5-CoV-2 evolution and infectivity.

**IMPORTANCE** HKU5-CoV-2 is a newly discovered bat coronavirus related to MERS-CoV that can infect human cells using the ACE2 receptor, raising concerns about possible transmission from animals to humans. This study provides the first proof that recombination occurs in HKU5-CoV-2 spike protein, which leads to the changes at position 498 in the RBD—which directly interacts with the human ACE2—and at position 722/723 and 729 in the FCS, may impact how efficiently the virus infects people. These mutations might help the virus adapt to animals or humans and spread more effectively. Our research shows that recombination is important in shaping HKU5-CoV-2's ability to infect bats and potentially humans.

**KEYWORDS** homologous recombination, coronavirus, linkage disequilibrium, merbecovirus, zoonotic spillover, spike protein, furin cleavage site

HKU5-CoV-2 merbecovirus is a newly discovered coronavirus from *Pipistrellus* spp. bats in China. Phylogenetically related to MERS-CoV, HKU5-CoV-2 can enter host cells via the ACE2 receptor present in many birds and mammals, including humans (1). Although there is no reported case of human transmission to date, HKU5-CoV-2 poses a risk of zoonotic spillover with pandemic potential. During viral replication, the genetic material is exchanged by different recombination events, which can contribute to genome diversity in RNA viruses (2). It is well known that intra- or inter-variant recombination quickly shapes the pathogenesis and evolution of coronaviruses, even within 3 weeks of the human transmission (3–6). However, whether recombination has influenced HKU5-CoV-2 infectivity or biological fitness is completely unknown. Here, we

Address correspondence to Ting-Yu Yeh, yehty@auxergen.com.

Vincent Tsai, Samuel M. Liao, Chia-En Hong, Feng-Yu Kuo, Yen Chun Wang, Michael C. Feehley, Patrick J. Feehley, Yi-Chen Lai, and Gregory P. Contreras contributed equally to this article.

T-.Y.Y. is the CSO and stock holder of Auxergen Inc and Auxergen s.r.l. G.P.C. is the CEO and stock holder of Auxergen Inc. and Auxergen s.r.l. The other authors have no conflict of financial interest.

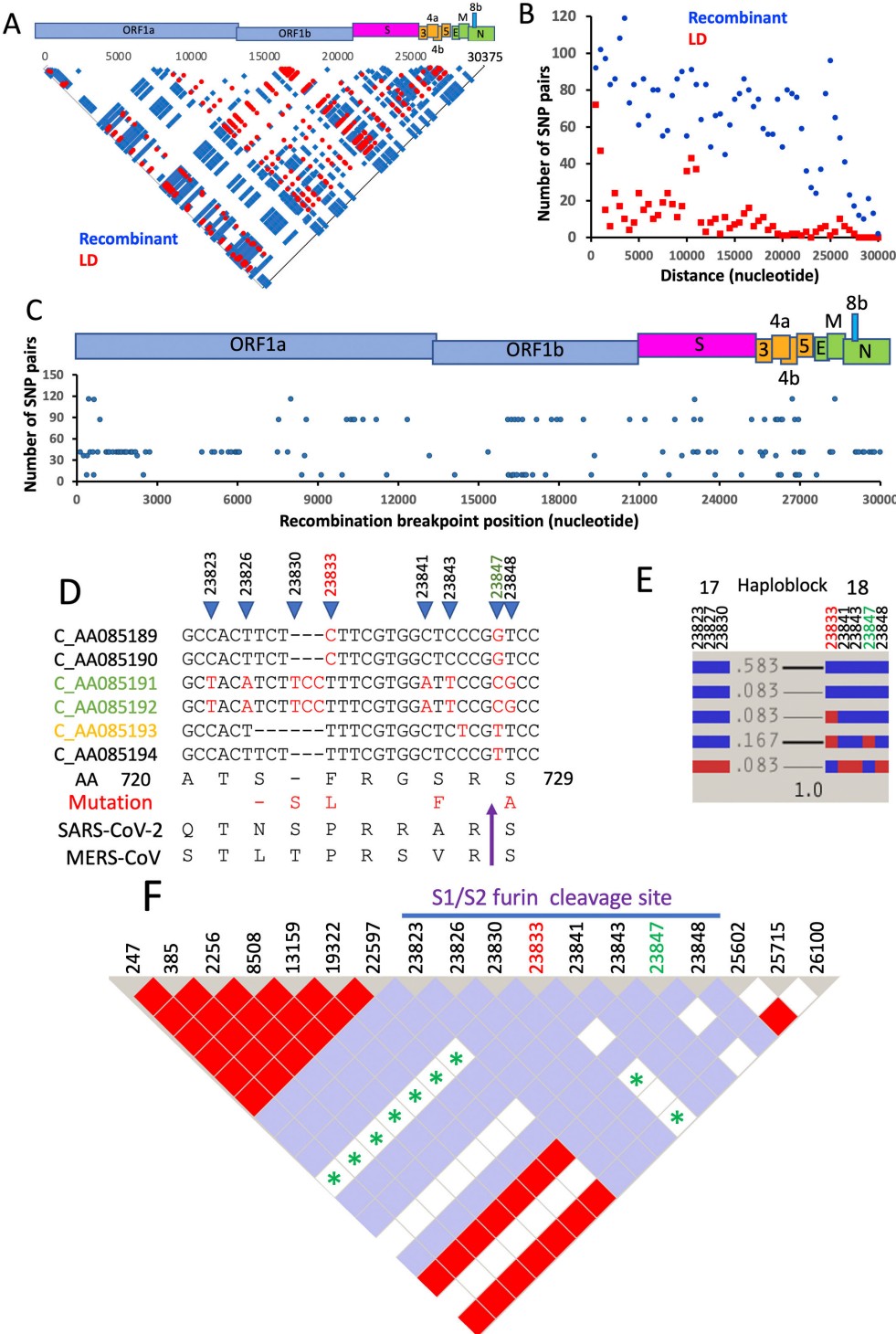

FIG 1 LD analysis of HKU5-CoV-2 genome. (A) Pairwise plots of LD (N = 666, red) and recombinant (N = 3896, blue) SNP pairs. LD is defined if pairs for which the value of logarithm of the odds (LOD) is above >2, the squared coefficient of correlation ($R^2$) = 1, and the high value of 95% confidence interval (CIhigh) bounds for $D'$ equal to 1. Recombination is defined if pairs for which the upper CI bound of $D'$ is below 0.9. (B) The nonrandom distribution of LD (red) and recombinant (blue) SNP pairs is shown by plotting the pair numbers (Y-axis) against the genomic distance between that pair (X-axis). (C) The positions of HKU5-CoV-2 recombination breakpoints, with the numbers of their associated recombinant SNP pairs, are shown in Y-axis. Genomic organization of HKU5-CoV-2 is shown above (A and C). (D–F) Recombination of HKU5-CoV-2 FCS. (D) Alignment of FCS sequences and recombination breakpoints SNP23833 and SNP23847. FCS (purple arrow), sequences with Ser272/273

**Fig 1 (Continued)**

deletion (yellow), or S729A substitution (green) are colored. (E) Haplotype block organization of (D). (F) HaploView LD map of FCS and other SNPs. Red squares indicate high levels of LD, and white and blue squares represent low levels of LD. Recombinant SNP pairs are marked with asterisks.

report the first evidence that recombination drives HKU5-CoV-2 diversity and has led to alterations in the receptor-binding domain (RBD) and S1/S2 furin cleavage site (FCS).

Six HKU5-CoV-2 sequences are available in GenBase (C-AA08189 to C-AA08194) and were analyzed by Haploview (1, 7) (Materials and Methods, Supplemental Information). Linkage disequilibrium (LD) analysis of 765 single-nucleotide polymorphisms (SNPs) shows the nonrandom feature of LD and recombination, with 666 LD and 3,896 recombinant SNP pairs (Fig. 1A and B). 167 recombination breakpoints and 27 haploblocks are present in the viral genome (Fig. 1C; Fig. S1; Table S5). 30 nonsynonymous mutations are associated with recombination breakpoints, including NSP2 (F278, C288, G309, S640, and G771), NSP3 (A1472, S1835, N1885, A1904, V1924, L2538, and T2957), the spike protein (H312, V498, V541, V642, F724, A729, and T1043), ORF3 (R21), ORF4a (D27, T29, S54, L91, N94, and R95), ORF4b (G155, G232), M protein (S51), and N protein (A192). Recombination in RNA viruses like coronaviruses occurs when the viral RNA-dependent RNA polymerase switches templates during replication in a co-infected cell. This process can generate a hybrid genome containing new combinations of SNP pairs from both parental viruses. However, single-nucleotide substitutions found in these recombinant HKU5-CoV-2 genomes are not necessarily caused by recombination itself, but are more likely the result of the inherently high error rate of the viral polymerase in parental genomes that contained substitutions before recombination happened.

ACE2 utilization has independently evolved multiple times among the HKU5 clade of merbecoviruses (8). Structural studies reveal that HKU5-CoV-2 has better adapted to human ACE2 than HKU5-CoV-1 (1, 8). Unlike betacoronaviruses (e.g., SARS-CoV-2), recombination breakpoints do not occur more frequently in the HKU5-CoV-2 spike gene (4.7/1,000 nucleotides) compared to NSP2 (14.6), ORF4a (32.6) and M (10.5) protein (9). However, we found that SNP23016/23043/23064/23156/23193/23285 in the RBD of the spike protein are breakpoints for intradomain or long-range recombination (Fig. 2A through C), indicating that the spike's binding interface with ACE2 (S447-D537) is a recombination hotspot (22.3/1,000 nucleotides). The breakpoints at SNP23156 and SNP23285 cause T498/V/I and V541A substitutions, respectively. It has been shown that T498 in HKU5-CoV-2-441 isolate (C_AA085194) directly contacts human ACE2 D30/H34/E37 and RBD G499/N503, which are also ACE2 binding residues (Fig. 2D through F) (1). *In silico* computer simulation analysis suggests that the V498 or I498 substitution loses hydrogen bonding with RBD G499/N503 as well as ACE2 E37. Hydrogen bond interaction is also missing in RBD N503S substitution (Fig. 2F). Taken together, these data suggest that recombination may change receptor binding, host range, and/or viral infectivity of HKU5-CoV-2. Further investigation of HKU5-CoV-2 virulence after gaining or losing beneficial residues like T498 and N503 (e.g., C_AA085191/192 vs C_AA085193/194) is needed for a more complete picture of viral evolution and transmission.

The FCS allows the coronavirus to enter human cells more efficiently. The PRRA tetrapeptide insertion to introduce the FCS ([680]SPRRAR↓SV[687], Fig. 1D) is considered a key determinant of virulence in the SARS-CoV-2 pandemic (10). It has been suggested that the FCS appeared independently multiple times in coronavirus evolution, but the FCS is also often lost during virus adaptation in the cell culture by either deletions or point mutations (11). We observed that the HKU5-CoV-2 FCS ([722]SSFRGSR↓S[729]) contains several mutations (S723del, S722/S723del, F724L, S727F, and S729A) (Fig. 1D). Haploblock analysis revealed that recombination occurred at SNP23830/23833 (Fig. 1E). Breakpoint SNP23833 and SNP23847 also form 9 and 41 recombinant SNP pairs, respectively,

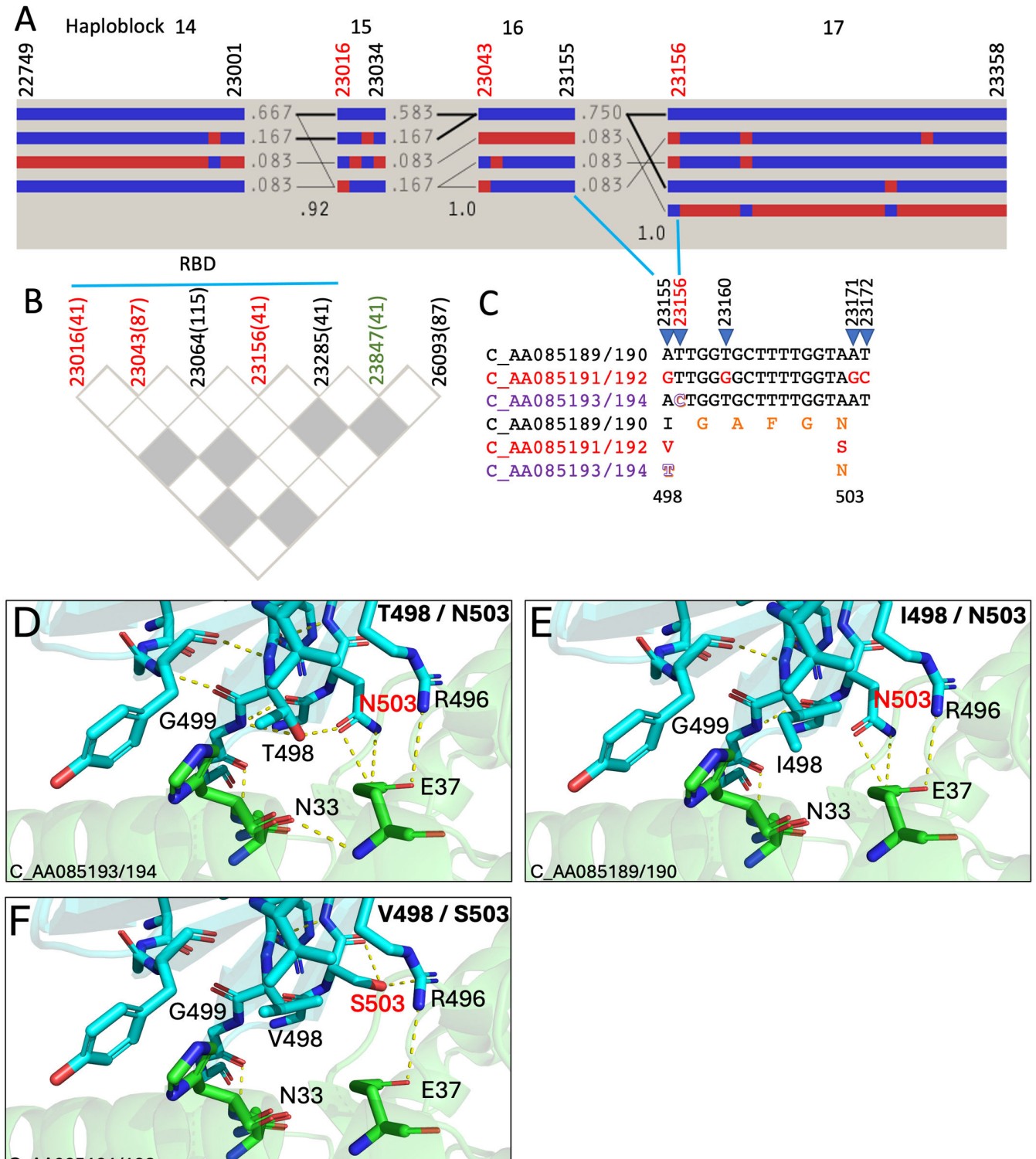

**FIG 2** Recombination of HKU5-CoV-2 RBD. (A) Haplotype block (14–17) organization of HKU5-CoV-2 RBD sequences. The Hedrick's multi-allelic *D*′ statistics in the crossing areas represent the recombination level between two haploblocks. Haplotype frequencies are shown as the numbers next to each haplotype block, or >10% (thick lines) or >1% (thin lines). The breakpoints in haploblocks 15, 16, and 17 are colored in red. (B) Pairwise LD plots of recombinant SNP pairs (white squares) in RBD, FCS (SNP23847, green), or the intergenic region between spike and E protein (SNP26093). The numbers of recombinant SNP pairs are associated with each SNP. Gray squares are neither LD nor recombinant SNP pairs (the upper CI bound of *D*′ is between 0.9 and 0.98). (C) Alignment of the adjacent sequences of SNP23156 breakpoints. ACE2 receptor binding residues are colored in orange. T498 is outlined (purple). (D–F) Residues with polar contacts within 5 Å of RBD residue 498. T498/N503 (D), I498/N503 (E), and V498/S503 (F) are illustrated by PyMOL (PDB: 9JJ6). RBD's ACE2 binding sites: G499, N503. ACE2 receptor's spike binding sites: E37. Hydrogen bonds are shown by yellow dashed lines. Rotamers with the lowest steric strain were chosen for mutated residues.

indicating that the FCS is a crossover hotspot (Fig. 1F and 2B). Breakpoint SNP23833 gives rise to S722/S723 deletion or insertion, while SNP23847 generates S729A or A729S substitution. Örd et al. reported that deletion of [679]NSPR of SARS-CoV-2 FCS abolishes the furin cleavage activity (12). S722 and S729 of HKU5-CoV-2 are homologous to S680 and S686 in SARS-CoV-2 FCS. SARS-CoV-2 S680 and S686 are phosphorylation sites for cyclin-dependent kinase and protein kinase A, respectively. Their phosphorylation significantly inhibits furin cleavage activity. Therefore, mutations of S722/S723 and S729 of the HKU5-CoV-2 spike protein could change furin cleavage activity (e.g., C_AA085189/190/194 vs C_AA085191/192/193).

Another non-LD algorithm, Recco (13), identified 10 putative recombination breakpoints in NSP4, NSP13, NSP14, and the transmembrane domain of the spike protein (Table S1). Other breakpoints (SNP18065/19205/25521) were also confirmed by LD and haploblock analysis. It is reasonable to speculate that recombination of HKU5-CoV-2 virus is underestimated.

Extremely common in the evolutionary history of SARS-like coronaviruses, recombination helps viruses acquire beneficial mutations that enhance virulence and transmission (2–5). Recombination can alter the host tropism of MERSr-CoVs and BtCoV-422 due to broader receptor usage (14). Viral polymerase can also make errors via slippage, which results in tandem duplication during template switching (15). Our findings show that these breakpoints create new mutations in HKU5-CoV-2 RBD and FCS, which could potentially affect viral entry and infectivity. Genomic surveillance of HKU5-CoV-2 and its recombinants in wild bat populations could be the key strategy to mitigate zoonotic transmission and future outbreaks.

## ACKNOWLEDGMENTS

T.-Y.Y. and G.P.C. had full access to all the data in the study and took responsibility for the integrity of the data and the accuracy of the data analysis. Concept and design: T.-Y.Y. Acquisition, analysis, or interpretation of data: T.-Y.Y., V.T., S.M.L., C.-E.H., F.-Y. K., Y.C.W., M.C.F., P.J.F., and Y.-C.L. Drafting of the manuscript: T.-Y.Y. and G.P.C. Critical revision of the manuscript for important intellectual content: T.-Y.Y. and G.P.C. Statistical analysis: T.-Y.Y., V.T., S.M.L., C.-E.H., F.-Y. K., Y.C.W., M.C.F., P.J.F., and Y.-C.L. Supervision: T.-Y.Y. and G.P.C. No AI is used for analysis, writing, or editing in this study.

## AUTHOR AFFILIATIONS

[1]Auxergen Inc. Rita Rossi Colwell Center, Baltimore, Maryland, USA
[2]Auxergen S.r.l., Tecnopolis Science and Technology Park of the University of Bari, University of Bari, Valenzano, Italy
[3]Taipei Wego Private Senior High School, Taipei City, Taiwan
[4]Davis Senior High School, Davis, California, USA
[5]Henry M. Gunn High School, Palo Alto, California, USA
[6]Taipei Municipal Chien Kuo High School, Taipei City, Taiwan
[7]Taipei Municipal Chenggong High School, Taipei City, Taiwan
[8]Department of Biophysics, Johns Hopkins University, Baltimore, Maryland, USA
[9]Department of Radiology, Taipei Veterans General Hospital, Taipei City, Taiwan
[10]School of Medicine, National Yang Ming Chiao Tung University, Taipei City, Taiwan

## AUTHOR ORCIDs

Ting-Yu Yeh http://orcid.org/0000-0001-5333-5486

## AUTHOR CONTRIBUTIONS

Ting-Yu Yeh, Conceptualization, Data curation, Formal analysis, Funding acquisition, Investigation, Methodology, Project administration, Resources, Software, Supervision, Validation, Visualization, Writing – original draft, Writing – review and editing | Vincent Tsai, Project administration, Writing – review and editing | Samuel M. Liao, Formal

analysis, Writing – review and editing | Chia-En Hong, Formal analysis, Writing – review and editing | Feng-Yu Kuo, Formal analysis, Writing – review and editing | Yen Chun Wang, Formal analysis, Writing – review and editing | Michael C. Feehley, Formal analysis, Writing – review and editing | Patrick J. Feehley, Formal analysis, Writing – review and editing | Yi-Chen Lai, Formal analysis, Writing – review and editing | Gregory P. Contreras, Formal analysis, Writing – review and editing

## ADDITIONAL FILES

The following material is available online.

### Supplemental Material

**Supplemental materials (Spectrum01420-25-s0001.docx).** Supplemental materials and methods, Fig. S1, and Tables S1.
**Table S3 (Spectrum01420-25-s0002.txt).** Information of recombination SNP pairs.
**Table S4 (Spectrum01420-25-s0003.rtf).** SNP numbers associated with the numbers above the haploblocks for Figure S1.
**Table S5 (Spectrum01420-25-s0004.txt).** Summary of recombination breakpoints in HKU5-CoV-2 in Figure 1C.

### Open Peer Review

**PEER REVIEW HISTORY (review-history.pdf).** An accounting of the reviewer comments and feedback.

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
