## [Reviewer comments · Microbiology Spectrum]

Microbiology Spectrum

Recombination alters the receptor binding and furin cleavage site in novel bat-borne HKU5-CoV-2 coronavirus

Ting-Yu Yeh, Vincent Tsai, Samuel Liao, Chia-En Hong, Feng-Yu Kuo, Yen Chun Wang, Michael Feehley, Patrick Feehley, Yi-Chen Lai, and Gregory Contreras

Corresponding Author(s): Ting-Yu Yeh, University of Maryland Institute of Marine and Environmental Technology

Review Timeline:

Submission Date:	May 8, 2025
Editorial Decision:	June 3, 2025
Revision Received:	August 7, 2025
Accepted:	August 10, 2025

Editor: Takamasa Ueno

Reviewer(s): The reviewers have opted to remain anonymous.

Transaction Report:

DOI: <https://doi.org/10.1128/spectrum.01420-25>

Re: Spectrum01420-25 (**Recombination alters the receptor binding and furin cleavage site in novel bat-borne HKU5-CoV-2 coronavirus**)

Dear Dr. Ting-Yu Yeh:

Thank you for the privilege of reviewing your work. Below you will find my comments, instructions from the Spectrum editorial office, and the reviewer comments.

Revision Guidelines

Sincerely,
Takamasa Ueno
Editor
Microbiology Spectrum

Reviewer #1 (Comments for the Author):

This brief report presents evidence that recombination shapes the receptor-binding domain (RBD) and the furin cleavage site (FCS) of the HKU5-CoV-2 spike protein, potentially altering its host range and zoonotic potential. The study is timely, given ongoing concerns regarding the emergence of novel zoonotic coronaviruses. The analysis based on linkage disequilibrium (LD), haplotype identification, and Recco recombination analysis is robust in concept and scope. I am adding some comments to

improve the overall quality of the manuscript:

- Authors indicate: 'Whether recombination has an influence on HKU5-CoV-2 is completely unclear to date'.

The sentence reads vague and unspecific. It is not specific in which parameter of the virus the recombination event has influence. I suppose authors refer to the influence of recombination in HKU5-CoV-2 infectivity or biological fitness. Please, clarify this.

- There is an inconsistent use of 'S1/S2 furin cleavage site' and 'FCS' across the manuscript. Please define the abbreviation the first time it appears in the text and use the abbreviation later.

- Authors indicate: 'These results suggest that, by joining two viral segments together, RNA recombination generated protein mutations (substitution, insertion, deletion) at breakpoints in HKU5-CoV-2 genome.'

The sentence gives the misleading impression that recombination involves a physical cutting and splicing of two viral genomes, as the authors describe it as "joining two segments together," which can be confusing. In reality, recombination in RNA viruses like coronaviruses occurs when the viral RNA-dependent RNA polymerase switches templates during replication in a co-infected cell. This process can generate a hybrid genome containing new combinations of SNPs pairs from both parental viruses (as authors are addressing in this manuscript). However, single-nucleotide substitutions found in the recombinant genome are not necessarily caused by recombination itself, but are more likely the result of the inherently high error rate of the viral polymerase in parental genomes that contained substitutions before recombination happened. Please, make this sentence more concise and clearer.

- Figure 2B caption, there is no indication about what gray squares

- Figures 1D-F could be clearer if a key color would be added, similar to 1A.

- AlphaFold3 / PyMOL version should be indicated. Any other relevant setting (including energy minimization) should also be mentioned.

Reviewer #2 (Comments for the Author):

In this observation, the authors found that HKU5-CoV-2 receptor binding domain and S1/S2 furin cleavage site in the spike protein are recombination hotspots and identified lots of recombination breakpoints and haploblocks. Some of the recombination may lead to the substitution of amino acid residue in RBD which directly contacts the ACE2 receptor and may cause deletion/insertion and substitution in the S1/S2 furin cleavage site. As a consequence, the host tropism and furin cleavage activity may be affected, raising the concerns about pandemic potential. Generally, this manuscript is well written. Below are some concerns.

1) The second paragraph in the main text: "Six HKU5-CoV-2 sequences are available in GenBase (C-AA08189 to CAA08194) and were analyzed by Haploview (1, 7, 8) (Appendix)." There are several "Appendix"s in the main text. There is no need using the word "Appendix", which is actually misleading. Just refer to the Materials and Methods section, References section or supplemental Figures and Tables in the Supplementary Information.

2) The fourth paragraph in the main text: "Breakpoint SNP23833 gives rise to S722/S723 deletion or insertion, while SNP23847 generates S729A or A729S substitution, both of which could change furin cleavage activity (e.g. C_AA085189/190/194 versus C_AA085191/192/193)." As the position of S722/S723 is a little bit far from the furin cleavage site and Serine and Alanine (S729A or A729S substitution after furin cleavage site) are both of small side chain, it is unclear to what extent the deletion/insertion and substitution will affect the furin cleavage activity. Could the author provide some supporting evidence about the impact of mutations at these two sites on furin cleavage efficiency which may be reported in previous literatures?

3) The last paragraph in the main text: "Viral polymerase makes errors via slippage, which results in tandem duplication during template switching (15)." Besides slippage, is there any other reason that can cause the recombination and mutation (substitution, deletion and insertion)? How about the lack of proof-reading function of viral RdRp?

4) The last paragraph in the main text: "Our findings show that these breakpoints create new mutations in HKU5-CoV-2 RBD and FCS, which could significantly affect viral entry and infectivity." Actually, due to the lack of experimental evidence or literature supporting in this manuscript, it is not very clear to what extent the mutations in RBD and FCS of HKU5-CoV-2 will affect the binding between the spike protein and ACE2 and the efficiency of furin cleavage on S1/S2. It is better to change the word 'significantly' to 'potentially'. Alternatively, could the author provide some information reported in literatures to support it, if experiments except for in silico computer simulation analysis cannot be conducted?

5) Figure 2C: The last code of C_AA085193/194 is AGC, which is translated to Ser. However, the corresponding amino acid residue is N in this Figure. Please check if there is any error in the amino acid sequence or in the nucleic acid sequence throughout the manuscript.

Dear reviewers:

We sincerely appreciate the reviewers' helpful and invaluable comments to improve the quality of our paper. Here are our point-to-point responses to the reviewers' questions:

Reviewer #1 (Comments for the Author):

This brief report presents evidence that recombination shapes the receptor-binding domain (RBD) and the furin cleavage site (FCS) of the HKU5-CoV-2 spike protein, potentially altering its host range and zoonotic potential. The study is timely, given ongoing concerns regarding the emergence of novel zoonotic coronaviruses. The analysis based on linkage disequilibrium (LD), haplotype identification, and Recco recombination analysis is robust in concept and scope. I am adding some comments to improve the overall quality of the manuscript:

Response: We thank that the reviewer considered our manuscript is robust and his/her effort to improve the quality of our paper.

- Authors indicate: 'Whether recombination has an influence on HKU5-CoV-2 is completely unclear to date'.

The sentence reads vague and unspecific. It is not specific in which parameter of the virus the recombination event has influence. I suppose authors refer to the influence of recombination in HKU5-CoV-2 infectivity or biological fitness. Please, clarify this.

Response: We thank to reviewer's comment. We changed the sentence as "Whether recombination has an influence on HKU5-CoV-2 infectivity or biological fitness is completely unclear to date."

- There is an inconsistent use of 'S1/S2 furin cleavage site' and 'FCS' across the manuscript. Please define the abbreviation the first time it appears in the text and use the abbreviation later.

Response: We labeled the first "S1/S2 furin cleavage site" as FCS and "receptor binding domain" as RBD the first time they appear in the text and use the abbreviation later.

- Authors indicate: 'These results suggest that, by joining two viral segments together, RNA recombination generated protein mutations (substitution, insertion, deletion) at breakpoints in HKU5-CoV-2 genome.'

The sentence gives the misleading impression that recombination involves a physical cutting and splicing of two viral genomes, as the authors describe it as "joining two segments together," which can be confusing. In reality, recombination in RNA viruses like coronaviruses occurs when the viral RNA-dependent RNA polymerase switches templates during replication in a co-infected cell. This process can generate a hybrid genome containing new combinations of SNPs pairs from both parental viruses (as authors are addressing in this manuscript). However, single-nucleotide substitutions found in the recombinant genome are not necessarily caused by recombination itself, but are more likely the result of the inherently high error rate of the viral polymerase in parental genomes that contained substitutions before recombination happened. Please, make this sentence more concise and clearer.

Response: We appreciate reviewer's insightful inputs. To avoid the confusion, we included reviewer's comment in this part :

“ Recombination in RNA viruses like coronaviruses occurs when the viral RNA-dependent RNA polymerase switches templates during replication in a co-infected cell. This process can generate a hybrid genome containing new combinations of SNPs pairs from both parental viruses. However, single-nucleotide substitutions found in these recombinant HKU5-CoV-2 genomes are not necessarily caused by recombination itself, but are more likely the result of the inherently high error rate of the viral polymerase in parental genomes that contained substitutions before recombination happened.”

- Figure 2B caption, there is no indication about what gray squares
- Figures 1D-F could be clearer if a key color would be added, similar to 1A.
- AlphaFold3 / PyMOL version should be indicated. Any other relevant setting (including energy minimization) should also be mentioned.

Response: (1) Figure 2B: add “Grey squares are neither LD nor recombinant SNP pairs (the upper CI bound of D' is between 0.9-0.98).” in Figure legend. (2) Because some SNPs (e.g. 23823, 23826, 23830, etc) listed here do not form recombinant SNPs (Figure 1F), so we did not label them as blue as in Figure 1A. (3) We added “In silico computer simulation of T498 and N503 substitution was

performed with AlphaFold 3.0.1 and presented by PyMOL (5, 6). Default settings for PyMOL version 3.1.3 were chosen. Besides automatically chosen minimal steric strain rotamers for substituted amino acids, no other energy minimizations were applied.” In Materials and Methods.

Reviewer #2 (Comments for the Author):

In this observation, the authors found that HKU5-CoV-2 receptor binding domain and S1/S2 furin cleavage site in the spike protein are recombination hotspots and identified lots of recombination breakpoints and haploblocks. Some of the recombination may lead to the substitution of amino acid residue in RBD which directly contacts the ACE2 receptor and may cause deletion/insertion and substitution in the S1/S2 furin cleavage site. As a consequence, the host tropism and furin cleavage activity may be affected, raising the concerns about pandemic potential. Generally, this manuscript is well written. Below are some concerns.

Response: We appreciate reviewer’s effort for inputs.

1) The second paragraph in the main text: "Six HKU5-CoV-2 sequences are available in GenBase (C-AA08189 to CAA08194) and were analyzed by Haploview (1, 7, 8) (Appendix)." There are several "Appendix"s in the main text. There is no need using the word "Appendix", which is actually misleading. Just refer to the Materials and Methods section, References section or supplemental Figures and Tables in the Supplementary Information.

Response: Thanks to reviewer’s comment. We removed all “Appendix” from the main text.

2) The fourth paragraph in the main text: "Breakpoint SNP23833 gives rise to S722/S723 deletion or insertion, while SNP23847 generates S729A or A729S substitution, both of which could change furin cleavage activity (e.g. C_AA085189/190/194 versus C_AA085191/192/193)." As the position of S722/S723 is a little bit far from the furin cleavage site and Serine and Alanine (S729A or A729S substitution after furin cleavage site) are both of small side chain, it is unclear to what extent the deletion/insertion and substitution will affect the furin cleavage activity. Could

the author provide some supporting evidence about the impact of mutations at these two sites on furin cleavage efficiency which may be reported in previous literatures?

Response: Thanks to reviewer's very valuable comments. We add the following sentences in the text:

“Örd et al. reported that deletion of 679NSPR of SARS-CoV-2 FCS abolishes the furin cleavage activity (Sci Rep. 2020 Oct 9;10:16944). S722 and S729 of HKU5-CoV-2 are homologous to S680 and S686 in SARS-CoV-2 FCS. SARS-CoV-2 S680 and S686 are phosphorylation sites for cyclin dependent kinase and protein kinase A, respectively. Their phosphorylation significantly inhibits furin cleavage activity. Therefore, mutations of S722/S723 and S729 of HKU5-CoV-2 spike protein could change furin cleavage activity (e.g. C_AA085189/190/194 versus C_AA085191/192/193).”

3) The last paragraph in the main text: "Viral polymerase makes errors via slippage, which results in tandem duplication during template switching (15)." Besides slippage, is there any other reason that can cause the recombination and mutation (substitution, deletion and insertion)? How about the lack of proof-reading function of viral RdRp?

Response: As mentioned by reviewer 1, we add the sentences (earlier in the text) to explain RNA recombination better. “Recombination in RNA viruses like coronaviruses occurs when the viral RNA-dependent RNA polymerase switches templates during replication in a co-infected cell. This process can generate a hybrid genome containing new combinations of SNPs pairs from both parental viruses. However, single-nucleotide substitutions found in these recombinant HKU5-CoV-2 genome are not necessarily caused by recombination itself, but are more likely the result of the inherently high error rate of the viral polymerase in parental genomes that contained substitutions before recombination happened.”

4) The last paragraph in the main text: "Our findings show that these breakpoints create new mutations in HKU5-CoV-2 RBD and FCS, which could significantly affect viral entry and infectivity." Actually, due to the lack of experimental evidence or literature supporting in this manuscript, it is not very clear to what extent the mutations in RBD

and FCS of HKU5-CoV-2 will affect the binding between the spike protein and ACE2 and the efficiency of furin cleavage on S1/S2. It is better to change the word 'significantly' to 'potentially'. Alternatively, could the author provide some information reported in literatures to support it, if experiments except for in silico computer simulation analysis cannot be conducted?

Response: We change “significantly” into “potential”. We have included the research of SARS-CoV-2 FCS to compare as described above.

5) Figure 2C: The last code of C_AA085193/194 is AGC, which is translated to Ser. However, the corresponding amino acid residue is N in this Figure. Please check if there is any error in the amino acid sequence or in the nucleic acid sequence throughout the manuscript.

Response: We really appreciate the reviewer’s comment. The sequence should be AAT, not AGC. We corrected C_AA085193/194 sequence in figure 2c. The Figure 2D is correct.

Re: Spectrum01420-25R1 (**Recombination alters the receptor binding and furin cleavage site in novel bat-borne HKU5-CoV-2 coronavirus**)

Dear Dr. Ting-Yu Yeh:

Your manuscript has been accepted, and I am forwarding it to the ASM production staff for publication. Your paper will first be checked to make sure all elements meet the technical requirements. ASM staff will contact you if anything needs to be revised before copyediting and production can begin. Otherwise, you will be notified when your proofs are ready to be viewed.

Sincerely,
Takamasa Ueno
Editor
Microbiology Spectrum

There were concerns that the English language usage in the manuscript might make it difficult to properly evaluate the science. The ASM Journals webpage provides links to various language editing services (<https://journals.asm.org/writing-your-paper#language-editing-services>). You may consider using these services when revising your manuscript. The use of these services will have no direct bearing on the editorial decision. ASM has no affiliation with these companies.